# Evaluation of Fosfomycin-Sulbactam Combination Therapy against Carbapenem-Resistant *Acinetobacter baumannii* Isolates in a Hollow-Fibre Infection Model

**DOI:** 10.3390/antibiotics11111578

**Published:** 2022-11-09

**Authors:** Sazlyna Mohd Sazlly Lim, Aaron Heffernan, Saiyuri Naicker, Steven Wallis, Jason A. Roberts, Fekade Bruck Sime

**Affiliations:** 1UQ Centre for Clinical Research, Faculty of Medicine, University of Queensland, Brisbane, QLD 4029, Australia; 2Department of Medicine, Faculty of Medicine and Health Sciences, Universiti Putra Malaysia, Serdang 43400, Malaysia; 3School of Medicine, Griffith University, Southport, QLD 4222, Australia; 4Departments of Pharmacy and Intensive Care Medicine, Royal Brisbane and Women’s Hospital, Brisbane, QLD 4029, Australia; 5Division of Anaesthesiology Critical Care Emergency and Pain Medicine, Nîmes University Hospital, University of Montpellier, 30900 Nîmes, France

**Keywords:** fosfomycin, sulbactam, combination, hollow-fibre infection model, *Acinetobacter baumannii*, carbapenem-resistant, pharmacodynamic

## Abstract

Static concentration in vitro studies have demonstrated that fosfomycin- or sulbactam-based combinations may be efficacious against carbapenem-resistant *Acinetobacter baumannii* (CRAB). In the present study, we aimed to evaluate the bacterial killing and resistance suppression potential of fosfomycin-sulbactam combination therapies against CRAB isolates in a dynamic infection model. We simulated clinically relevant dosing regimens of fosfomycin (8 g every 8 h, 1 h infusion) and sulbactam (12 g continuous infusion or 4 g every 8 h, 4 h infusion) alone and in combination for 7 days in a hollow-fibre infection model (HFIM) against three clinical isolates of CRAB. The simulated pharmacokinetic profiles in the HFIM were based on fosfomycin and sulbactam data from critically ill patients. Fosfomycin monotherapy resulted in limited bacterial killing. Sulbactam monotherapies resulted in ~ 3 to 4 log_10_ kill within the first 8 to 32 h followed by regrowth of up to 8 to 10 log_10_ CFU/mL. A combination of fosfomycin and continuous infusion of sulbactam led to a ~2 to 4 log_10_ reduction in bacterial burden within the first 24 h, which was sustained throughout the duration of the experiments. A combination of fosfomycin and extended infusion of sulbactam produced a ~4 log_10_ reduction in colony count within 24 h. This study demonstrated that fosfomycin in combination with sulbactam is a promising option for the treatment of MDR *A. baumannii*. Further studies are needed to further assess the potential clinical utility of this combination.

## 1. Introduction

*Acinetobacter baumannii* is a difficult pathogen to effectively treat as it has the ability to protect itself in the face of antibiotic therapy via various mechanisms of resistance [1]. Carbapenems have previously been the agent of choice to counteract species producing extended spectrum beta-lactamase (ESBL), including *A. baumannii*. However, carbapenems are losing viability for the treatment of *A. baumannii* infections as the prevalence of carbapenem-resistant *A. baumannii* (CRAB) increases [2]. Furthermore, many of the newer antibiotics under development or available on the pharmaceutical market for drug-resistant Gram-negative bacilli are active against KPC-producers or carbapenem-resistant *P. aeruginosa* but not against CRAB [3]. Except for cefiderocol, other newer agents lack activity against *A. baumannii* [4].

Considering the limited antibiotics that are available for the treatment of CRAB infections and the paucity of new agents that are active against CRAB strains, antibiotic combination therapy appears to be an appealing treatment alternative. Most antibiotic combination studies for the treatment of multidrug-resistant (MDR) *A. baumannii* have looked at polymyxin-based combinations [4]. However, the use of polymyxin is often limited by its narrow therapeutic window and nephrotoxicity [5], making polymyxin-based combinations less than ideal. An ideal antibiotic combination would be one that is active against the bacteria of interest, able to suppress the emergence of resistance and has a better safety profile.

Fosfomycin- or sulbactam-based combinations may be considered as prospective treatment substitutes against CRAB. High-dose (≥4 g/day) sulbactam-based combinations have been shown to result in better clinical improvement and clinical cure compared to colistin-based combinations in the treatment of MDR *A. baumannii* [6]. In terms of the safety profile of high-dose sulbactam, studies have demonstrated that sulbactam doses of 9–12 g/day can be used safely in a clinical setting [7,8].

Numerous fosfomycin- and sulbactam-based combinations have been studied in vitro at fixed concentrations [9,10,11]. Synergism has been observed in various fosfomycin- and sulbactam-based combinations against MDR *A. baumannii* with synergy rates ranging from 30 to 80% [12,13,14,15]. In one of our earlier reports, we explored the fosfomycin and sulbactam combination in checkerboard and static time-kill studies [16]. Two-drug combination checkerboard assays demonstrated that the fosfomycin-sulbactam combination was synergistic against 37/50 CRAB (74%) isolates. This finding was further supported by static time-kill experiments, which demonstrated that the combination was synergistic at fosfomycin and sulbactam concentrations of 128 mg/L, respectively, for two out of two isolates tested.

Despite compelling data seen in static in vitro infection models, the role of combination therapy such as fosfomycin and sulbactam in a clinical setting remains ambiguous, as the dynamicity of the in vitro models and in vivo conditions clearly differs. Therefore, in the present study, we set out to evaluate the combination of fosfomycin and sulbactam against CRAB isolates in a dynamic infection model. We assessed the bacterial killing activity and suppression of resistance of the fosfomycin and sulbactam combination at clinically relevant plasma exposures.

## 2. Materials and Methods

### 2.1. Bacterial Isolates

Fifty isolates were obtained from The University of Queensland Centre of Clinical Research (UQCCR). These isolates were chosen from samples previously analysed by Zowawi et al. [17]. These 50 isolates were used to screen for synergistic activity against various antibiotic combinations, including the fosfomycin and sulbactam combination, in a previous study [16]. Out of the fifty isolates, three isolates were chosen for the hollow-fibre infection model (HFIM) experiments.

The isolates were stored in cation-adjusted Mueller-Hinton II broth (CA-MH) (Becton, Dickinson and Company, Sparks, MD, USA) containing 20% glycerol in a freezer at −80 °C. Before each experiment, fresh isolates were streaked out and grown on Mueller-Hinton agar (Becton, Dickinson and Company, Sparks, MD, USA) plates, incubated at 37 °C for 18 h and used for the preparation of the inoculum. Carbapenem resistance was determined by Zowawi et al. [17] using disk diffusion susceptibility testing for imipenem (10 μg) and meropenem (10 μg) following the European Committee on Antimicrobial Susceptibility Testing (EUCAST) methodology.

### 2.2. Antimicrobial Agents

Meropenem (Tokyo Chemical Industry Co. Ltd., Tokyo, Japan), sulbactam (Acros Organics, Lot: A0405260, Shanghai, China) and fosfomycin (Wako Pure Chemical Industries, Ltd., Osaka, Japan) were obtained from their respective manufacturers. Stock solutions of these antibiotics were prepared in sterile Milli-Q water, filter-sterilised with a 0.22 µm polyvinylidene difluoride (PVDF) syringe filter, aliquoted and stored at −80 °C until required. Before each susceptibility test, an aliquot of the drug was thawed and diluted to the desired concentrations with cation-adjusted Mueller-Hinton II (CA-MH) broth. Broth or agar containing fosfomycin was supplemented with 25 mg/L of glucose-6-phosphate (Sigma-Aldrich, St. Louis, MO, USA).

### 2.3. In Vitro Susceptibility

The MICs of meropenem and sulbactam against the three CRAB isolates were determined by the broth microdilution method, in quadruplicate, following the recommendations of CLSI, as described in the CLSI M100 approved standard [18]. The mode MIC was reported for each isolate. Fosfomycin susceptibility testing was performed by agar dilution [18]. *Klebsiella pneumoniae* (ATCC 700603), *Pseudomonas aeruginosa* (ATCC 27853) and *Enterococcus faecalis* (ATCC 29212) strains were used as quality control strains for sulbactam, fosfomycin and meropenem, respectively. The fosfomycin and sulbactam MIC of the isolates from all the experimental arms were also reassessed at the end of the experiment using the same methods.

### 2.4. Hollow-Fibre Infection Model

The HFIM was put together using FiberCell Systems cellulosic cartridges (C3008) in a 37 °C incubator as detailed previously [19]. In the experiments investigating the fosfomycin-sulbactam combination, a supplementing compartment was introduced to simulate the differential clearance of the two antibiotics as described by Blaser [20]. The CA-MH broth used in all experiments was supplemented with 25 mg/L glucose-6-phosphate. All experiments were conducted over seven days. One HFIM experiment was conducted for each dosing regimen and isolate combination. For each isolate, an overnight bacterial suspension was prepared in CA-MH broth at 36 °C and subsequently diluted to achieve an initial inoculum of approximately 6.5 log_10_ CFU/mL.

Sulbactam exhibits primarily time-dependent bacterial killing [21] and has been shown to be 93% stable for 24 h at 37 °C [22]. Therefore, a continuous or extended infusion of sulbactam could maximise pharmacodynamic target attainment. For fosfomycin, reports on the pharmacodynamic index associated with bacterial killing are mixed. Some studies have reported that fosfomycin exhibits time-dependent activity against *Staphylococcus aureus* and *Escherichia coli* [23,24]. Other more recent studies on fosfomycin demonstrated concentration-dependent bacterial killing against *E. coli* [25,26]. In the present study, considering concentration-dependent activity and the ratio of the area under the concentration-time curve to the MIC (AUC/MIC) as the driver of efficacy for fosfomycin, an intermittent infusion at a high dose was chosen to maximise this index. The following treatment regimens were simulated in the HFIM experiments: (1) fosfomycin 8 g every 8 h, given as a 1 h infusion (intermittent infusion); (2) sulbactam 12 g daily, given as a continuous infusion (preceded by a 4 g loading dose, given as a 1 h infusion); (3) sulbactam 4 g every 8 h, given as a 4 h infusion (extended infusion). These fosfomycin and sulbactam regimens were simulated alone and in combination in the HFIM. A loading dose of sulbactam (4 g, 1 h infusion) was given prior to the continuous infusion of sulbactam to rapidly achieve steady-state concentration. A no-treatment growth control was also included.

The simulated pharmacokinetic profiles in the HFIM were based on the pharmacokinetic models of fosfomycin and sulbactam developed for critically ill patients [27,28]. To ensure the simulated profiles are representative of the central location of the target profile, a normal patient body weight of 70 kg and a creatinine clearance of 100 mL/min were used in the covariate models for the estimation of primary PK parameters. For fosfomycin, negligible protein binding [29], a 4 h elimination half-life [27] and peak concentration of 300 mg/L were used to simulate free-drug exposures. For sulbactam, 38% protein binding [30], a 1.5 h elimination half-life [28] and peak concentration of 60 mg/L were used to simulate free-drug exposures. The simulations were deemed acceptable if the best-fit peak concentrations and elimination half-lives were both within 20% of the target values [31]. A one-compartment model was fitted to the observed sulbactam and fosfomycin concentration-time profiles using the Pmetrics package (v1.52, Laboratory of Applied Pharmacokinetics and Bioinformatics, Los Angeles, CA, USA) for R (v.3.6.2, R Core Team, Vienna, Austria).

The clearance of each drug was simulated by controlling the flow rates of the CA-MH broth flowing through the inner lumens of the hollow-fibre cartridge. Peristaltic pumps (Masterflex L/S; Cole-Parmer, Vernon Hills, IL, USA) were used to provide CA-MH broth flow through the central reservoir. A Duet Pump (FiberCell Systems Inc., New Market, MD, USA) was used to provide continuous circulation between the central and peripheral compartments to ensure continuous media and drug distribution.

### 2.5. Quantification of Viable Bacterial Populations

Bacterial quantification was performed with periodic sampling at 0, 2, 4, 6, 8, 24, 32, 48, 72, 96, 120, 144 and 168 h from the cartridge’s extracapillary space sampling port. Samples were washed twice in phosphate-buffered saline to minimise antibiotic carry-over. A 100 μL aliquot of an appropriately diluted bacterial suspension was manually plated onto CA-MH agar. The limit of quantification was 2-log_10_ CFU/mL.

### 2.6. Fosfomycin and Sulbactam Assays for Pharmacokinetics

Samples were taken from the central compartment sampling port at 1, 2, 4, 8, 24, 25, 28, 32, 48, 49, 52, 56, 72, 96, 120, 144 and 168 h for pharmacokinetic analysis to ascertain whether the intended pharmacokinetic profiles of the antibiotics were achieved in the HFIM. These samples were immediately stored at −80 °C until bioanalysis.

Concentrations of fosfomycin and sulbactam in CA-MH broth were measured over a calibration range of 1 to 1000 mg/L by a validated ultra-high performance liquid chromatography using a tandem mass spectrometry (UHPLC-MS/MS) method on a Nexera2 UHPLC system coupled with a 8030+ triple quadrupole mass spectrometer (Shimadzu, Kyoto, Japan). Test samples were assayed alongside plasma calibrators and quality controls and met the batch acceptance criteria (US FDA 2018).

For fosfomycin, the CA-MH broth (20 μL) was spiked with internal standard (ethylphosphonic acid), vortex mixed and centrifuged. The supernatant was injected into the UHPLC-MS/MS instrument. The stationary phase was a SeQuant zic-HILIC 2.1 × 50 mm (5.0 μm) analytical column protected by a 20 mm SeQuant zic-HILIC guard cartridge (Merck, Darmstadt, Germany) operated at room temperature. Mobile phase A was 1 mM ammonium acetate (pH 4.5), and mobile phase B was 100% acetonitrile with 0.2% formic acid (*v*/*v*). The method was isocratic (25% mobile phase B). Fosfomycin was monitored by negative mode electrospray at m/z of 137.0→79.0. Ethylphosphonic acid was monitored in negative mode at 109.05→79.0. The lower limit of quantitation (LLOQ) was 1 mg/L. Inter-batch precision was 4.4, 6.2 and 7.0% at 5, 50 and 5000 mg/L, respectively. Inter-batch accuracy was 5.5, 2.1 and −4.4% at 5, 50 and 5000 mg/L, respectively.

For sulbactam, the CA-MH broth sample (10 μL) was spiked with internal standard (tazobactam) and protein was precipitated with acetonitrile. An aliquot of 1 μL of the supernatant was injected into the UHPLC-MS/MS. The stationary phase was Shim-pack XR-ODS III 2 × 50 mm (1.6 µm) analytical column (Shimadzu, Kyoto, Japan) preceded by a C18 UHPLC analytical guard column (Phenomenex, Torrence, CA, USA). Mobile phase A was 0.1% formic acid (*v*/*v*), and mobile phase B was 100% acetonitrile with 0.1% formic acid (*v*/*v*). The mobile phase was delivered as a gradient from 10% B to 75% B and back again at a flow 0.35 mL/min, producing a backpressure of approximately 8500 psi. Ionisation was performed by electrospray. Sulbactam was monitored in negative mode at 232.05→140.05. The internal standard tazobactam was monitored in negative mode at 299.00→138.15. The assay method was linear from 0.78 to 156 mg/L (LLOQ of 0.78 mg/L). Inter-assay precision was 6.7, 1.5 and 13.6% and accuracy was 1.7, −7.0 and −10.8% at 2.7, 18 and 144 mg/L sulbactam in CA-MH broth, respectively.

## 3. Results

### 3.1. In Vitro Susceptibility

The fosfomycin MIC for isolates #75, #98 and #102 was 128, 256 and 128 mg/L, respectively. The sulbactam MIC was 64 mg/L for all three isolates. These isolates were confirmed to be carbapenem-resistant with a meropenem MIC ranging between 32 to 64 mg/L.

### 3.2. Hollow-Fibre Infection Model

The fosfomycin and sulbactam pharmacokinetic profiles were reasonably well simulated in the HFIM. Typical profiles for fosfomycin and sulbactam regimens are shown in Figure 1. The predicted versus observed pharmacokinetic profiles of 8 g fosfomycin every 8 h (1 h infusion), 4 g sulbactam loading dose (1 h infusion) and 12 g continuous infusion every 24 h and 4 g sulbactam every 8 h (4 h infusion) showed reasonable agreement with r^2^ values of 0.89, 0.92 and 0.92, respectively. Changes in viable counts are shown in Figure 2. The fosfomycin monotherapy resulted in limited bacterial killing against isolates #75 and #102 (stasis within the first 4 to 8 h, followed by growth up to ~10 log_10_ CFU/mL). Against isolate #98, fosfomycin monotherapy led to a 3 log_10_ reduction in bacterial burden by 8 h, followed by regrowth. The sulbactam continuous infusion in monotherapy resulted in ~3 to 4 log_10_ kill within the first 8 to 32 h followed by regrowth up to 8 to 10 log_10_ CFU/mL. The sulbactam extended infusion regimen yielded a ~3 to 4 log_10_ kill by 8 to 32 h after the start of infusion, with subsequent regrowth of up to 10 log_10_ CFU/mL.

The combination of fosfomycin and continuous infusion of sulbactam led to a ~2 to 4 log_10_ reduction in the bacterial burden within the first 24 h, which was sustained throughout the duration of the experiments for isolates #75 and #102. The combination of fosfomycin and extended infusion of sulbactam, on the other hand, produced a ~4 log_10_ reduction in colony count within 24 h followed by suppression of regrowth.

### 3.3. In Vitro Susceptibility Post-Drug Exposure

Table 1 summarises the fosfomycin and sulbactam MICs post exposure. Exposure to fosfomycin monotherapy led to an increase in the fosfomycin MIC by 3 to 4 fold; whereas, the sulbactam MIC remained the same. A 2 fold increase in the sulbactam MIC was observed after exposure to sulbactam monotherapies; whereas, the fosfomycin MIC remained the same despite exposure to sulbactam. Exposure to the combination of fosfomycin and continuous infusion of sulbactam did not affect the subsequent sulbactam MIC of the isolates. We did, however, observe a 2-fold rise in the fosfomycin MIC. The post-exposure in vitro susceptibility testing for the fosfomycin and extended infusion of sulbactam combination arms were not performed as there was no colony growth on the CA-MH agar.

## 4. Discussion

To the best of our knowledge, this is the first study to explore the in vitro activity of fosfomycin-sulbactam combinations against CRAB in a dynamic infection model. The most intriguing finding in this study is the extent of bactericidal activity observed in the fosfomycin and sulbactam extended infusion arms against the tested isolates, despite the inherent resistance of *A. baumannii* to fosfomycin [32]. Similarly, a HFIM study by Lenhard et al. [33] demonstrated that high-dose ampicillin/sulbactam (sulbactam total daily dose 12 g/day) in combination with meropenem and polymyxin B resulted in rapid clearance of a carbapenem- and polymyxin-resistant *A. baumannii* isolates. These findings support the role of sulbactam in the treatment of MDR *A. baumannii*.

Several studies have also demonstrated benefits of fosfomycin-based combination therapy in patients with MDR *A. baumannii* infection. Sirijatuphat et al. [34] observed better microbiological eradication within 72 h of treatment (90.7% vs. 58.1%, *p* = 0.001) and a trend toward better clinical outcomes and mortality rates in the colistin-fosfomycin combination group compared with the colistin monotherapy group. Furthermore, a prospective, observational, multicentre study demonstrated that fosfomycin-containing combinations were associated with 30-day survival (hazard ratio 0.04, CI 95% 0.01–0.13, *p* = 0.001) in patients with severe pneumonia due to MDR *A. baumannii* [35]. Intravenous fosfomycin has also been shown to demonstrate a favourable safety profile, with only mild adverse effects, such as mild hypokalaemia due to high sodium load with administration of fosfomycin disodium, which did not require discontinuation of therapy [36,37]. Similarly, several studies have shown that sulbactam doses of 9–12 g/day can be used safely in a clinical setting [7,8].

As our understanding of the molecular mechanism driving sulbactam’s activity against *A. baumannii* is still not complete, we are not able to fully explain the mechanism of fosfomycin-sulbactam synergy. However, the synergy is likely due to the inhibition of bacterial cell wall synthesis at two different stages in the biosynthesis pathway. Fosfomycin binds to phosphoenolpyruvate transferase thereby inhibiting mucopeptide synthesis, which is an early step in the synthesis peptidoglycan polymer that makes the cell wall [38]. It therefore acts at an earlier stage of bacterial cell wall production than most other antibiotics that inhibit cell wall synthesis [38]. Sulbactam is a class A beta-lactamase inhibitor with intrinsic whole-cell activity against certain bacterial species, including *A. baumannii* [39]. Sulbactam binds to penicillin-binding proteins of *Acinetobacter* spp., subsequently affecting a later stage of bacterial cell wall biosynthesis involving peptidoglycan cross-linkage. It is hypothesised that this activity accounts for the bactericidal effects of sulbactam on *A. baumannii* [40].

Another notable finding is the superiority of the sulbactam extended infusion regimen compared to the sulbactam continuous infusion regimen in terms of bacterial killing and suppression of resistance against CRAB isolates seen in this study. Neither the sulbactam extended infusion nor the continuous infusion regimens achieved concentrations that exceeded the sulbactam MIC of 64 mg/L (Figure 1). Nonetheless, the sulbactam extended infusion regimen performed better, probably due to the relatively higher concentration it achieved (C_max_ ~50 mg/L), compared to the continuous infusion regimen (steady-state concentration 25–30 mg/L). Lenhard et al. [33] demonstrated that an ampicillin/sulbactam dosing regimen of 8/4 g every 8 h, when simulated in the HFIM, was able to maintain a percentage of time above the MIC of 74% for sulbactam against *A. baumannii* isolates with sulbactam MIC 16 mg/L, which likely explained the drastic killing observed during ampicillin/sulbactam monotherapy in their study. However, considering the high sulbactam MIC of the CRAB isolates used in the present study, the sulbactam extended infusion was not able to maintain a decent percentage of time above the MIC, as the simulated dosing regimen did not achieve concentrations that exceeded the sulbactam MIC of 64 mg/L (Figure 1). Nonetheless, the combination containing the sulbactam extended infusion was still able to exhibit bacterial killing and suppression of regrowth.

It was expected that the CRAB isolates tested would develop further resistance to fosfomycin and sulbactam following exposure to fosfomycin and sulbactam monotherapy [41]. Interestingly, we observed an increase in fosfomycin MIC, but not sulbactam MIC, following exposure to the combination therapy, in which sulbactam was given as a continuous infusion. As combination regimens containing 4 g sulbactam every 8 h (extended infusion) and 12 g continuous infusion exhibited differing abilities to prevent further fosfomycin resistance, we hypothesised that this observation was likely due to the difference in the sulbactam concentration at steady state between the two regimens (sulbactam extended infusion vs. continuous infusion). It appears that sulbactam is able to prevent the further development of fosfomycin resistance in these isolates, in a concentration-dependent manner. The exact mechanism by which this phenomenon occurs should be studied further.

In this study, we used CRAB isolates that have high MICs (fosfomycin MIC 128–256 mg/L and sulbactam MIC 64 mg/L) to mirror difficult-to-treat, worst-case scenarios. The sulbactam MIC of the isolates used in our HFIM experiments were much higher compared with the MICs reported in other studies [42,43]. For fosfomycin, studies have shown that 70–80% of CRAB isolates have fosfomycin MICs of 128–256 mg/L [13,44]. This finding is supported by our own study, where we found that 66% of our CRAB isolates had similar fosfomycin MICs of 128–256 mg/L [45].

The dosing regimens simulated were chosen on the basis of our in silico study [46], which demonstrated the highest probability of target attainment of 2 log_10_ kill (71.6%) when doses of fosfomycin 24 g/day and sulbactam 12 g/day were used in combination. Attainment of 2 log_10_ kill is important, as discussed by Drusano et. al. [47], as it is this magnitude of bacterial burden reduction that is associated with near-maximal granulocyte-mediated bacterial kill. Total daily doses were limited to those used in a clinical setting as the goal was to identify optimal regimens for quick assimilation into clinical practice. Additionally, these HFIM experiments represent a conservative estimate of bactericidal activity as this infection model lacks an immune response and therefore does not account for the effect of the native immune system.

Nonetheless, this study is not without its limitations. Firstly, our study was performed against three clinical CRAB isolates, and therefore the findings might not be applicable to all MDR *A. baumannii* isolates. However, as observed in our previous checkerboard study, the fosfomycin-sulbactam combination is synergistic in a good fraction of CRAB isolates (~70%) [16]; therefore, the current findings are likely indicative of microbiological outcomes in a large proportion of CRAB isolates. Future studies should be carried out using MDR *A. baumannii* isolates with varying fosfomycin and sulbactam MICs to confirm these findings.

Secondly, whilst we acknowledge that the quantification of emergent resistant strains using agar plates containing the antibiotic at concentrations several times higher than the MIC is a common way to explore effects on resistant subpopulations, this was not feasible in our study as the isolates used were already resistant with high fosfomycin and sulbactam MICs. However, we have assessed the MIC of fosfomycin and sulbactam post-treatment to identify any shift from the baseline. Thirdly, we were unable to explain the difference in the response to the treatment arms between the three isolates. It has been discussed previously [4] that there is limited understanding to date as to what type of strains are more susceptible to combination therapy. Indeed, this may be a significant reason for the lack of reproducibility of results from in vitro studies.

Additionally, we only tested dosing regimens requiring 24 g/day of fosfomycin and 12 g/day of sulbactam, albeit these regimens are clinically accepted. Further studies, in which the doses of fosfomycin and sulbactam are titrated to establish a dose-response curve, are required to fully inform rational dosing of this combination. The time-kill profile of the fosfomycin and sulbactam combination presented in this study may also vary at different sites of infection, particularly considering the differences in site-specific pharmacokinetics in humans. Further studies in controlled clinical trials are needed to evaluate the use of this combination for the treatment of CRAB infections.

## 5. Conclusions

In conclusion, this study demonstrated that fosfomycin in combination with sulbactam is a promising option in our fight against MDR *A. baumannii*. We found that fosfomycin in combination with extended infusion of sulbactam was the most active of all the regimens evaluated against three CRAB isolates in a seven-day HFIM. The bacterial clearance observed in the HFIM suggests that clinicians may be able to prevent the emergence of resistance during therapy through dosing regimen modifications and the strategic selection of combination regimens.

## Figures and Tables

**Figure 1 antibiotics-11-01578-f001:**
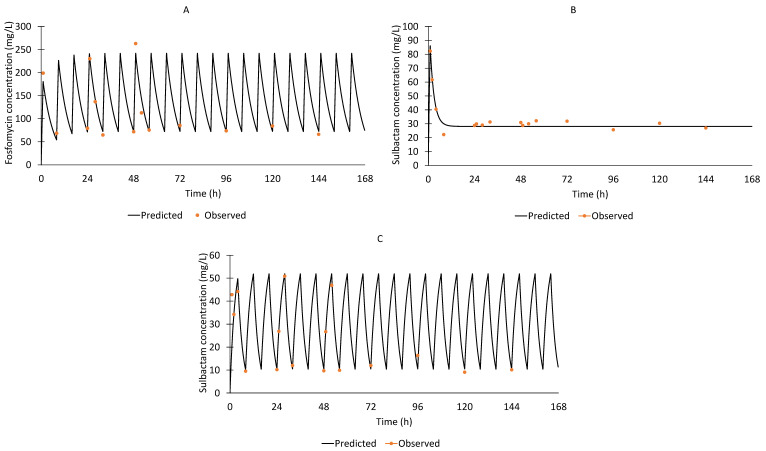
Typical simulated pharmacokinetic profile of (**A**) 8 g fosfomycin every 8 h (1 h infusion) (r^2^ = 0.89); (**B**) 4 g sulbactam loading dose (1 h infusion) and 12 g continuous infusion every 24 h (r^2^ = 0.92); (**C**) 4 g sulbactam every 8 h (4 h infusion) (r^2^ = 0.92).

**Figure 2 antibiotics-11-01578-f002:**
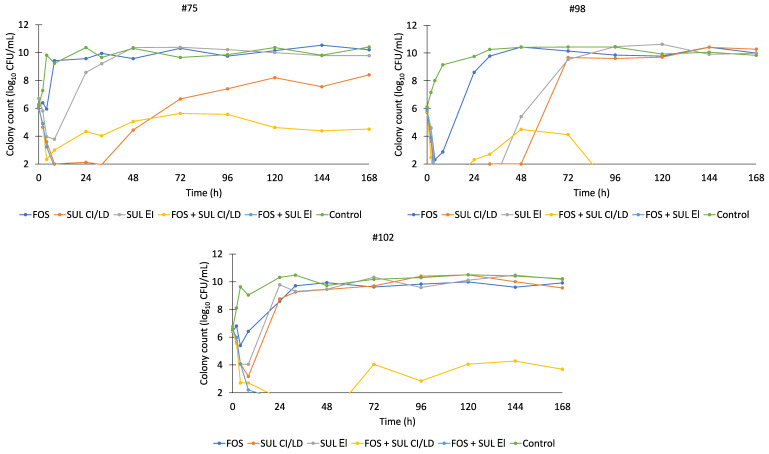
Time-kill curves for isolates #75, #98 and #102. FOS, fosfomycin; SUL, sulbactam; EI, extended infusion; CI, continuous infusion; LD, loading dose.

**Table 1 antibiotics-11-01578-t001:** Fosfomycin and sulbactam minimum inhibitory concentration post-antibiotic exposure.

Isolates	Pre-Exposure MIC (mg/L)		Antibiotic Regimen	Post-Exposure MIC (mg/L)	
	Fosfomycin	Sulbactam		Fosfomycin	Sulbactam
#75	128	64	Fosfomycin monotherapy	2048	32
			Sulbactam CI monotherapy	128	64
			Sulbactam EI monotherapy	128	256
			Fosfomycin + Sulbactam CI	512	32
			Fosfomycin + Sulbactam EI	-	-
#98	256	64	Fosfomycin monotherapy	2048	64
			Sulbactam CI monotherapy	256	128
			Sulbactam EI monotherapy	128	128
			Fosfomycin + Sulbactam CI	-	-
			Fosfomycin + Sulbactam EI	-	-
#102	256	64	Fosfomycin monotherapy	1024	32
			Sulbactam CI monotherapy	128	256
			Sulbactam EI monotherapy	128	256
			Fosfomycin + Sulbactam CI	512	32
			Fosfomycin + Sulbactam EI	-	-

MIC, minimum inhibitory concentration; CI, continuous infusion; EI, extended infusion.

## Data Availability

Not applicable.

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
