# Peer review of "Evaluation of Fosfomycin-Sulbactam Combination Therapy against Carbapenem-Resistant Acinetobacter baumannii Isolates in a Hollow-Fibre Infection Model"

_antibiotics, 2022, doi:10.3390/antibiotics11111578_

Round 1
Reviewer 1 Report
Please add in methods a description of how the statistical analysis were conducted
Author Response
Please add in methods a description of how the statistical analysis were conducted
Response: We thank the reviewer for this comment. No statistical analysis for the singlicate runs customary in hollow fiber infection model studies. We and several other authors have published, in highly reputable journals, a number of hollow fiber studies with singlicate runs. Few examples are Journal of Antimicrobial Chemotherapy, 77 (11), 3026-3034, Antimicrobial Agents and Chemotherapy, 66 (9), e0016222; International Journal of Antimicrobial Agents, 60 (2) 106623, 1-7.; JOURNAL OF ANTIMICROBIAL CHEMOTHERAPY. doi:10.1093/jac/dkac323 etc. It is customary in hollow fibre infection studies to preferably use multiple isolates instead of replicates of the same isolates. Therefore, statistical analysis was not required.
Reviewer 2 Report
The study “Evaluation of fosfomycin-sulbactam combination therapy against carbapenem-resistant Acinetobacter baumannii isolates in a hollow-fibre infection model” form Lim S et al. explores an interesting topic. But I order in order to increase the usefulness and significance of the study, it needs a revision before being considered suitable for readers and there are some points to overcome for acceptance.
Authors divided abstract in 3 sections as per journal guidelines please reform it.
In this manuscript, author evaluated the bacterial killing and resistance suppression potential of fosfomycin-sulbactam combination against CRAB isolates in a dynamic infection model. However, the introduction of these sections is quite simple, not in deep. It is recommended to tone up the introduction section.
Too small replication: According to M&M only the in vitro susceptibility, Hollow-Fibre infection model and quantification viable bacterial population experiments were performed only once. It is recommended to perform twice with 3 replications at least.
Statistical analysis with pair-wise testing using the Student’s t-test need to be performed, and significant changes indicate by asterisks and also need to state what error on graphs. I would also suggest plotting individual data points on graphs instead of/alongside the means as this increases transparency and gives the reader a clearer idea of whether the assumptions made by statistical analysis. Additionally, you could use a Kruskal-Wallis test followed by Mann-Whitney tests adjusted for multiple comparisons to compare with untreated control.
It is suggested a moderate English revision by an English native speaker in order to polish text from typos and imperfections.
Unwanted spacing and typo mistakes throughout the manuscript. Need to be check and correct carefully.
Double check the way of adding references in the main text body and reference section as per journal guidelines.
Why author chose only 3 isolates out of 50? Please explain.
Please use consistent style for all units throughout the manuscript. E.g., Use h or hour.
Please remove "-"(Soft hyphen) after numbers.
Please use consistent style for “Minus sign” Line 88 and Line 167.
In figure 2 represent Y axis scale up to zero.
Author Response
The study “Evaluation of fosfomycin-sulbactam combination therapy against carbapenem-resistant Acinetobacter baumannii isolates in a hollow-fibre infection model” form Lim S et al. explores an interesting topic. But I order in order to increase the usefulness and significance of the study, it needs a revision before being considered suitable for readers and there are some points to overcome for acceptance.
Authors divided abstract in 3 sections as per journal guidelines please reform it.
Response: Thank you for pointing this out. This has been revised.
In this manuscript, author evaluated the bacterial killing and resistance suppression potential of fosfomycin-sulbactam combination against CRAB isolates in a dynamic infection model. However, the introduction of these sections is quite simple, not in deep. It is recommended to tone up the introduction section.
Response: We thank the reviewer for this comment. However, we prefer to provide a succinct and brief background avoid extensive literature review given this is an original study. If the editor strongly feels about expanding the background literature review, we are happy to expand this as needed.
Too small replication: According to M&M only the in vitro susceptibility, Hollow-Fibre infection model and quantification viable bacterial population experiments were performed only once. It is recommended to perform twice with 3 replications at least.
Response: We thank the reviewer for this comment and agree that this is a good approach. Our approach has also been used in other published studies such as Grimella (2020) and Onufrak (2020). Additionally, to stay within available research budget, we were unfortunately unable to run the HFIM twice for each isolate and treatment arms. As discussed above, we opted to run in multiple isolates instead of replicates of the same isolates.
Statistical analysis with pair-wise testing using the Student’s t-test need to be performed, and significant changes indicate by asterisks and also need to state what error on graphs. I would also suggest plotting individual data points on graphs instead of/alongside the means as this increases transparency and gives the reader a clearer idea of whether the assumptions made by statistical analysis. Additionally, you could use a Kruskal-Wallis test followed by Mann-Whitney tests adjusted for multiple comparisons to compare with untreated control.
Response: We thank the reviewer for this comment. Please see the response to the same comment above under reviewer 1.
It is suggested a moderate English revision by an English native speaker in order to polish text from typos and imperfections.
Response: Upon reviewer’s suggestion, the manuscript has been spell-checked again.
Unwanted spacing and typo mistakes throughout the manuscript. Need to be check and correct carefully.
Response: This has been checked again, upon reviewer’s comment.
Double check the way of adding references in the main text body and reference section as per journal guidelines.
Response: This has been done.
Why author chose only 3 isolates out of 50? Please explain.
Response: To stay within available research budget, it was decided that 3 isolates would be an acceptable number of isolates to be tested for each combination. For each combination, we performed HFIM for 3 isolates with 6 treatment/control arms for each isolates, making the number of arms for 1 combination to 18 arms in total. The study was performed not only for this combination but 1 other combination, making it a total of 36 arms. We have also acknowledged in the Discussion section that further studies should be carried out using other MDR A. baumannii isolates with varying fosfomycin and sulbactam MICs to add further evidence.
Please use consistent style for all units throughout the manuscript. E.g., Use h or hour.
Response: Thank you for pointing this out. This has been corrected.
Please remove "-"(Soft hyphen) after numbers.
Response: This has been corrected.
Please use consistent style for “Minus sign” Line 88 and Line 167.
Response: Thank you for pointing this out. This has been corrected.
In figure 2 represent Y axis scale up to zero.
Response: The limit of quantification was 2-log10 CFU/mL, hence why the Y axis scale is up to 2.
Reviewer 3 Report
In this manuscript, the authors mainly tested the effectiveness of treating three clinical isolates of carbapenem-resistant A. baumannii using a combination of fosfomycin and sulbactam in a 7-day HFIM with dosing regimens requiring 24 g/day of fosfomycin and 12 g/day of sulbactam, showing that fosfomycin in combination with sulbactam could be promising for treating some PDRAB.
1. It would add to the manuscript if the authors could discuss the advantages and limitations of choosing to test the combination of fosfomycin and sulbactam in HFIM.
2. What does the author mean by saying a dynamic model in line 18 while using “in a hollow-fibre infection model” in the title? How should HFIM be a dynamic model? I suggest authors articulate this in the introduction section.
3. Line 23 mentioned that “ fosfomycin monotherapy resulted in limited bacterial Killing”, why would that be the case? Due to its limited bacterial killing, why would the authors persist in testing the effectiveness of fosfomycin combining with sulbactam? And why can the efficacy of fosfomycin be improved when in combination with sulbactam?
4. Line 29, why would the authors be confident that ‘’fosfomycin in combination with sulbactam is a promising option for treatment of MDR A. baumannii’’, without in the research comparing its effectiveness with a positive control like cefiderocol or combination of comparator agents of fosfomycin?
5. Line 36, it seems more accurate to replace “treat” with “effectively treat”.
6. Line 44, since cefiderocol can act actively against A. baumannii, why not try screening agents that might have similar structures as cefiderocol or modifying cefiderocol to achieve better effectiveness but instead try to test the efficacy of the combination of agents?
7. Line 55, the author wrote: fosfomycin- or sulbactam-based combinations may be considered as prospective treatment substitutes against CRAB. However, the authors only explained why sulbactam-based combinations would be a better choice but did not suggest why fosfomycin combined with sulbactam might be a prospective treatment.
8. It would be better if the authors could specify in the manuscript how many log10 reduction of A. baumannii should be considered clinically relevant.
9. In Figure 1, the author did not make it clear where the predicted curve comes from. I suggest more observations between time points 72 and 96, 96 and 120, 120 and 144, and 144 and 168 should be implemented.
10. In Figure 2, the author did not do technical replicates of each A. baumannii strain and a statistical test of the result is needed to confirm that the result shown is significant and not due to serendipity. The authors should discuss why there exist differences when treating the three strains of A. baumannii using the same treatment.
11. Besides assessing the potential clinical utility of this combination, would this combination produce potential toxicity to the human body? It would be great if the authors could discuss that in the discussion.
12. In line 196, “m” behind 6.7 should be deleted.
13. In line 204, why did the author say that “The fosfomycin and sulbactam pharmacokinetic profiles were reasonably well simulated in the HFIM"? Please provide more explanation.
Author Response
In this manuscript, the authors mainly tested the effectiveness of treating three clinical isolates of carbapenem-resistant A. baumannii using a combination of fosfomycin and sulbactam in a 7-day HFIM with dosing regimens requiring 24 g/day of fosfomycin and 12 g/day of sulbactam, showing that fosfomycin in combination with sulbactam could be promising for treating some PDRAB.
- It would add to the manuscript if the authors could discuss the advantages and limitations of choosing to test the combination of fosfomycin and sulbactam in HFIM.
Response: This has been described in the Introduction and Discussion sections of the manuscript. (lines 66-81, 279-295)
- What does the author mean by saying a dynamic model in line 18 while using “in a hollow-fibre infection model” in the title? How should HFIM be a dynamic model? I suggest authors articulate this in the introduction section.
Response: The HFIM has been described in detail in the Methodology section (lines 124-172)
- Line 23 mentioned that “ fosfomycin monotherapy resulted in limited bacterial Killing”, why would that be the case? Due to its limited bacterial killing, why would the authors persist in testing the effectiveness of fosfomycin combining with sulbactam? And why can the efficacy of fosfomycin be improved when in combination with sulbactam?
Response: It is known that A. baumannii is inherently resistant to fosfomycin. However, the beauty of antibiotic synergy is that such resistance can sometimes be overcomed, as seen in this study. However, the mechanisms by which this occurs is not fully understood. We have attempted to delineate the possible mechanism in lines 296-308.
We were interested in this combination due to the findings that were seen in our checkerboard and TKA studies performed prior to the HFIM study (lines 75-81).
- Line 29, why would the authors be confident that ‘’fosfomycin in combination with sulbactam is a promising option for treatment of MDR A. baumannii’’, without in the research comparing its effectiveness with a positive control like cefiderocol or combination of comparator agents of fosfomycin?
Response: The conclusion was based on microbiological findings observed from the HFIM studies. The combination of fosfomycin and continuous infusion of sulbactam led to a ~ 2- to 4-log10 reduction in the bacterial burden within the first 24 h, which was sustained throughout the duration of the experiments. Combination of fosfomycin and extended infusion of sulbactam produced a ~ 4-log10 reduction in colony count within 24 h. These findings suggest that there is potential there. And it is not within the scope of this study to compare with a novel agent such as cefiderocol.
- Line 36, it seems more accurate to replace “treat” with “effectively treat”.
Response: Thank you for pointing this out. It has been revised as suggested.
- Line 44, since cefiderocol can act actively against A. baumannii, why not try screening agents that might have similar structures as cefiderocol or modifying cefiderocol to achieve better effectiveness but instead try to test the efficacy of the combination of agents?
Response: Thank you for the comment. Whilst we agree that the suggestion is good and relevant, it is out of the scope of the PhD project of which this study is a part of.
- Line 55, the author wrote: fosfomycin- or sulbactam-based combinations may be considered as prospective treatment substitutes against CRAB. However, the authors only explained why sulbactam-based combinations would be a better choice but did not suggest why fosfomycin combined with sulbactam might be a prospective treatment.
Response: There is limited studies looking at fosfomycin-sulbactam combination specifically. We have included lines 75-81 (from our earlier study) to provide additional support.
- It would be better if the authors could specify in the manuscript how many log10 reduction of A. baumannii should be considered clinically relevant.
Response: Per EMA guidelines on the use of PK/PD in the development of antimicrobials, the minimum analyses should report the PK/PD targets for achieving net bacterial stasis and 1- and 2-log10 reductions in bacterial densities. Additionally, as discussed by Drusano et. al. (2014), at least a 2-log10 kill is required for near-maximal granulocyte-mediated bacterial cell kill.
Upon reviewer comment, lines 346-349 have been added to clarify this.
Lines 346-349: Attainment of 2-log10 kill is important, as discussed by Drusano et. al. [48], as it is this magnitude of bacterial burden reduction that is associated with near-maximal granulocyte-mediated bacterial kill.
- In Figure 1, the author did not make it clear where the predicted curve comes from. I suggest more observations between time points 72 and 96, 96 and 120, 120 and 144, and 144 and 168 should be implemented.
Response: Thank you for pointing this out. The predicted curves were generated using Pmetrics package for R. This has been further clarified in lines 163-166.
Lines 163-166: A one-compartment model was fitted to the observed sulbactam and fosfomycin concentration-time profiles using Pmetrics package (v1.52, Laboratory of Applied Pharmacokinetics and Bioinformatics, Los Angeles, California) for R (v.3.6.2, R Core Team, Vienna, Austria).
Whilst we appreciate the reviewer comments and agree that additional timepoints would be useful, we are unable to re-run the HFIM models to include these additional PK timepoints, as this study was part of a PhD project which is now completed.
- In Figure 2, the author did not do technical replicates of each A. baumannii strain and a statistical test of the result is needed to confirm that the result shown is significant and not due to serendipity. The authors should discuss why there exist differences when treating the three strains of A. baumannii using the same treatment.
Response: Regarding replicates, please see the response above for the same comment.
With regards to the difference in response, it is something that is still not well understood. Upon reviewer comment, this has been discussed as a limitation to the study (lines 367-371).
Lines 367-371: Thirdly, we were unable to explain the difference in the response to the treatment arms between the three isolates. It has been discussed previously [4] that there is limited understanding to date as to what type of strains are more susceptible to combination therapy. Indeed, this may be a significant reason to the lack of reproducibility of results from in vitro studies
- Besides assessing the potential clinical utility of this combination, would this combination produce potential toxicity to the human body? It would be great if the authors could discuss that in the discussion.
Response: Thank you for the comment. The safety profile of fosfomycin has been described in lines 286-293 in the Discussion section.
Upon reviewer suggestion we have added the following lines to reflect the safety profile of sulbactam:
Lines 293-295: Similarly, several studies have shown that sulbactam doses of 9-12 g/day can be used safely in the clinical setting [7, 8].
- In line 196, “m” behind 6.7 should be deleted.
Response: Thank you for pointing this out, this has been deleted.
- In line 204, why did the author say that “The fosfomycin and sulbactam pharmacokinetic profiles were reasonably well simulated in the HFIM"? Please provide more explanation.
Response: The statement was based on Figure 1 which showed the predicted vs observed pharmacokinetic profile of the regimens of fosfomycin and sulbactam used in the study. To further clarify the following lines 224-227 have been added to the manuscript:
Lines 224-227: The predicted versus observed pharmacokinetic profiles of 8 g fosfomycin every 8 h (1h-infusion), 4 g sulbactam loading dose (1h-infusion) and 12 g continuous infusion every 24 h and 4 g sulbactam every 8 h (4h-infusion) showed reasonable agreement with r2 values of 0.89, 0.92 and 0.92, respectively.